# Information Processing Using Networks of Chemical Oscillators

**DOI:** 10.3390/e24081054

**Published:** 2022-07-31

**Authors:** Jerzy Gorecki

**Affiliations:** Institute of Physical Chemistry, Polish Academy of Sciences, Kasprzaka 44/52, 01-224 Warsaw, Poland; jgorecki@ichf.edu.pl

**Keywords:** chemical computing, network, oscillators, top-down design, Oregonator model, Japanese flag problem

## Abstract

I believe the computing potential of systems with chemical reactions has not yet been fully explored. The most common approach to chemical computing is based on implementation of logic gates. However, it does not seem practical because the lifetime of such gates is short, and communication between gates requires precise adjustment. The maximum computational efficiency of a chemical medium is achieved if the information is processed in parallel by different parts of it. In this paper, I review the idea of computing with coupled chemical oscillators and give arguments for the efficiency of such an approach. I discuss how to input information and how to read out the result of network computation. I describe the idea of top-down optimization of computing networks. As an example, I consider a small network of three coupled chemical oscillators designed to differentiate the white from the red points of the Japanese flag. My results are based on computer simulations with the standard two-variable Oregonator model of the oscillatory Belousov–Zhabotinsky reaction. An optimized network of three interacting oscillators can recognize the color of a randomly selected point with >98% accuracy. The presented ideas can be helpful for the experimental realization of fully functional chemical computing networks.

## 1. Introduction

Chemical computers are everywhere because all living organisms use them for acquiring and transmitting information and for decision-making. Animals and humans, using chemical computers represented by their nervous systems and brains [1], are able to control complex life processes such as orientation in space, navigation in crowded environments, creating models of the environment they live in, developing self-awareness and even predicting the future. This demonstrates that Nature-made chemical computers can perform very complex computational tasks with low energy consumption.

The information-processing industry is based on semiconductor technology. The unprecedented success of this technology in machine information processing [2] is possible because semiconductor logic gates are highly reliable. They are characterized by a long time of error-free operation and can be downsized to the nanoscale [3]. The gates can be concatenated within a single integrated circuit and perform more complex information processing functions. These properties of semiconductor information-processing devices perfectly match the bottom-up design strategy, according to which more complex operations are represented by combinations of simpler tasks for which constructions of corresponding circuits have already been developed [4].

The concept of logic gates and binary information coding, so successfully used for information processing with semiconductor devices, has strongly influenced other fields of unconventional computation, including the use of chemicals [5]. There are many reports on chemical realization of logic gates or binary operations [6,7,8,9,10,11,12,13,14]. The fact that a chemical medium allows for implementation of all basic logic gates proves that a universal chemical computer can be constructed. In exceptional cases, molecular logic gates used as molecular probes offer an interesting alternative to standard techniques [14]. However, most chemical logic gates, especially those constructed with a reaction-diffusion medium [15], are neither small nor fast. For the Belousov–Zhabotinsky (BZ) reaction, the output signal of a gate appears a few seconds after the input is introduced [7,8,10]. For other chemical media, this time can be much longer. In the case of information coded in DNA molecules, it may take a few hours before the gate answer is obtained [16]. In typical experimental conditions, the time of reliable chemical gate operation is measured in hours, not years as for semiconductors. In my opinion, it would be difficult to make a chemical device in which many chemical gates are concatenated, communicate and demonstrate stable functionality for a long time.

I think the BZ reaction [17,18] is the most frequently used medium in experimental studies on chemical computation. The BZ reaction is a complex catalytic oxidation of an organic substrate (usually malonic acid) in an acidic environment [19,20]. Two stages of the BZ reaction can be visually identified. One is fast oxidation of the catalyst, and the other is slow reduction of the catalyst by an organic substrate. The solution color reflects concentrations of catalyst in the oxidized and reduced forms. Therefore, many types of nonlinear evolution of the medium as oscillations or spatio-temporal patterns can be easily observed. If the BZ reaction proceeds in a spatially distributed medium, then local excitation corresponding to a high concentration of the reaction activator can propagate in space in the form of a concentration pulse. This type of behavior resembles the propagation of nerve impulses in living organisms. As a result, the BZ reaction has attracted attention as a medium for experiments with neuron-like chemical computing [21,22,23].

Within the most popular approach to computing with BZ medium, it is assumed that information is transmitted by propagating pulses of the oxidized form of catalyst. For binary coding, the presence of a pulse represents the logic TRUE state, and the state with a low concentration of the catalyst in the oxidized form is the logic FALSE state [13,21]. If the ruthenium complex ($Ru{\left(bpy\right)}_{3}^{2+}$) is used as the reaction catalyst, then the BZ reaction becomes photosensitive [24,25] and can be externally controlled by illumination. Oscillations can be inhibited by light, which means that for the same initial concentrations of reagents, the medium oscillates in the dark, is excitable at a low light intensity, and shows a strongly attracting stable steady-state when illumination is strong. In a medium with the photosensitive BZ reaction, excitable channels through which signals can propagate can be formed by specific illumination of a spatially distributed medium. Using a suitable geometry of excitable and non-excitable channels, one can control the propagation of excitations and, for example, make a signal diode [26], a memory cell [27,28] or logic gates [13,15]. However, in typical applications, such gates are big (with an area of about 1 cm${}^{2}$), and a single operation takes more than 10 s [20]. Therefore, the bottom-up approach from gates to complex information processing tasks does not look promising if binary information coding is used with BZ medium.

Literature reports demonstrate that a chemical medium can be efficiently applied for specific computing tasks. Parallel processing of information by the medium as a whole is a common feature of efficient implementations. The classic example is the Adleman experiment proving that the Hamiltonian path problem can be solved with DNA molecules [29,30]. Another example is the so-called prairie-fire algorithm for verifying if there is a path linking two randomly selected points in a labyrinth. This problem can be solved by a labyrinth formed from an excitable medium where stable pulses of excitation can propagate [31,32]. If there is a path linking two points, an excitation generated at one of the points will then appear at the other, and the time difference between excitation and detection can be used to estimate the shortest path linking these points. Yet another famous computing application of a chemical medium working in parallel is the image processing of black and white photos performed using a photosensitive variant of the BZ reaction proceeding in a uniform, spatially distributed system [33,34]. In such a medium, image processing is the consequence of a non-homogeneous initial state generated by initial illumination with intensity proportional to the grayscale of pixels of the processed image. In all methods mentioned above, the output information is coded in the time evolution of the computing medium.

However, the number of man-written parallel algorithms that can be efficiently executed using a chemical medium of parallel algorithms for chemical computers is very limited. A top-down design strategy offers a promising approach for finding new ones. The strategy can be summarized as follows. In the beginning, we select a problem we want to solve and the computing medium that is supposed to do it. Next, we define how the input information is introduced and how the output is extracted from observing medium evolution. The top-down approach can be applied if the properties of the medium—and thus of the medium evolution—can be controlled by a number of adjustable parameters. Within this strategy, we are supposed to find the values of parameters for which the medium answer (the output) gives the most accurate solution to the considered problem. To perform such optimization, we need a number of examples (the training dataset) that can be used to verify the accuracy of computation performed by the medium.

Networks of interacting chemical oscillators seem to be an interesting candidate for a chemical computing medium. Networks of BZ oscillators can be assembled using droplets containing reagents stabilized by lipids dissolved in the surrounding oil phase [35,36,37]. The idea of information processing with networks of interacting chemical reactors was formulated in [38]. In such an approach, a node is defined by a set of reactions sharing the specific reagents. Interaction between nodes consists of reactions providing the exchange of reagents between nodes. In [38,39,40,41], the authors focused on nodes that show excitable or bistable behavior; thus, the concentration of reagents in a single node can evolve towards one of two values that can be interpreted as corresponding to binary logical values [6]. It has been demonstrated that such nodes can perform logic gate operations. The idea of computing with oscillator networks generalizes the approach described above. First, the dynamics of a node are more complex. Second, the node answer is not a stationary state but reflects the character of evolution observed within a finite time interval. As the system is continuously evolving, the time of observation is an important parameter.

Several theoretical studies demonstrate the computing potential of oscillator networks [42,43,44]. It has been shown that oscillator networks could be optimized to perform classification tasks [45,46] and process information with the best possible use of the chemical medium. In this paper, I am concerned with the previously reported determination of color for a randomly selected point on the Japanese flag [47]. I introduce a few new elements to computing oscillator networks, including the comparison between two Oregonator models for node evolution that exhibit the different character of oscillations and generalization of the node–node interaction model with coupling parameters individually adjusted for pairs of nodes. Moreover, a new concept of extracting the output information from the network is proposed. In all previous papers on the subject, the number of activator maxima observed on a selected network node was regarded as the network answer. Here the total amount [48] of activator or inhibitor observed on the output oscillator is regarded as the network output. The results presented below demonstrate that such an approach is equally useful and leads to similar accuracy in determining the color of a point on the flag based on its coordinates as the number of activator maxima.

The paper is organized as follows. The information on the computational problem I am concerned with, the mathematical model of the time evolution of a node and of the network, and the optimization procedure are described in Section 2. Section 3 contains obtained results and their discussion. The final section presents verification of obtained results and presents suggestions for future studies.

## 2. Information Processing with Oscillator Networks

In this section, I present general information on the types of problems that can be solved with oscillator networks. Moreover, I discuss the network structure and a chemical model used to simulate its evolution.

### 2.1. Classification Type Problems

In a number of recent papers, it has been shown that networks of interacting chemical oscillators can be trained to perform classification tasks with reasonable accuracy [45,46,49]. Let us consider a problem *A* defined by a set of records $D_{A}=\left\{r_{n},n=1,N\right\}$. Each record $r_{n}=\left(p_{n}^{1},p_{n}^{2},\cdots ,p_{n}^{k},s_{n}\right)$ is in the form of a (k+1) tuple, where the first *k* elements are predictors represented by real numbers, and the last element ($s_{n}$) is the record type, and it is represented by an integer. It is assumed that the number of possible predictor values is limited. Let $D_{A}$ denote a database of records related to problem *A*. The classifier of $D_{A}$ is supposed to return the correct data type if the predictor values are used as its input.

There are classification problems for which $D_{A}$ is finite. For example the AND logic gate is equivalent to classification of the database: $D_{A}=\left\{\left\{0,0,0\right\},\left\{1,0,0\right\},\left\{0,1,0\right\},\left\{1,1,1\right\}\right\}$. In this paper, I consider a geometrically inspired problem of determining the color of a randomly selected point located on the Japanese flag (cf. Figure 1). The red disk (sun) is centrally located in a white square (here represented by the Cartesian product [−0.5,0.5]×[−0.5,0.5]). Let us notice that the location of the Japanese flag differs from the one considered in our previous paper on the problem [47] where it was [0.0,1.0]×[0.0,1.0]. This has been done intentionally to see if object location can influence classifier accuracy. The disk radius is $r=1/\sqrt{(}2\pi )$; thus, the areas of the sun and the white region are equal. Records of the considered problem have the form: (x,y,s), where (x,y)∈[−0.5,0.5]×[−0.5,0.5] are the point coordinates, and the record type s∈{0,1} tells if the point (x,y) is located in the red s=1 or in the white region s=0. A network that gives a random answer or a network that always gives the same answer (“the point is red” or “the point is white”) to all inputs solves the problem with 50% accuracy (or with a 50% chance of obtaining the wrong answer). I postulate that the Japanese flag problem can be solved with much higher accuracy by a network of chemical oscillators.

### 2.2. The Node Model

Before applying the top-down network optimization strategy, we should select the medium that is supposed to perform the classification. Here I use the two-variable Oregonator model [50] of the BZ reaction to describe the time evolution of an individual oscillator. The model equations are:(1)$$\begin{array}{ccc} \frac{du}{dt} & = & \frac{1}{\varepsilon }\left(u\left(t\right)-u{\left(t\right)}^{2}-\left(fv\left(t\right)+\phi \left(t\right)\right)\frac{u(t)-q}{u(t)+q}\right)-\alpha u\left(t\right) \end{array}$$(2)$$\begin{array}{ccc} \frac{dv}{dt} & = & u(t)-v(t) \end{array}$$
where *u* and *v*, respectively, represent concentrations of the reaction activator *U* corresponding to $HBrO_{2}$ and inhibitor *V* that in the two-variable Oregonator model is the oxidized form of the catalyst. The time evolution of a medium where the BZ reaction proceeds is determined by the values of parameters: f,q and ε. The parameter ε sets up the time scale ratio between variables *u* and *v*, *q* is the scaling constant, and *f* is the stoichiometric coefficient. The time-dependent function ϕ(t) that appears in Equation (Equation 1) is related to medium illumination. The Oregonator model is computationally simple, and it allows for performing complex evolutionary optimization involving the massive number of evaluations of network evolution needed for evolutionary optimization. It takes into account the effect of the combined excitation of one node by a few neighbors. Despite its simplicity, the Oregonator model provides a better-than-qualitative description of many phenomena related to the BZ reaction. It correctly describes the oscillation period as a function of reagent concentration and also can be used to simulate nontrivial phenomena such as the migration of a spiral in an electric field [51] or reaction of a propagating pulse to time-dependent illumination [52]. Of course, a model with a larger number of variables gives a more realistic description of the BZ reaction but, on the other hand, requires a more precise model of interactions between oscillators.

The last term in Equation (Equation 1) describes the activator decay, and it does not appear in the standard form of the Oregonator model. This term can be related to a reaction:(3)U+D→products
where *D* describes the other reagents of the process that occur with the rate α. As I discuss later, the existence of this process is equivalent to the presence of a “sink” node in the network that adsorbs the activator molecules.

The reported simulations of network evolution have been performed for two different sets of Oregonator model parameters. The time evolution in one of the considered networks is described by Model I: ε=0.2, q=0.0002 and f=1.1, which was used in the previous study on the color of a point on the Japanese flag [47]. For this set, the period of oscillations is ∼10.7 time unit if ϕ(t)=0. The optimization of other networks was done for the Oregonator Model II defined by: ε=0.3, q=0.002 and f=1.1. The period of oscillations is ∼8 time unit for ϕ(t)=0. The character of activator and inhibitor oscillations for the above-mentioned models and α=0.7 or α=0.5 is illustrated in Figure 2. For both sets of parameters, the system converges to a stable stationary state for ϕ∼0.2 [25,53].

The value of ϕ can be interpreted as the light intensity in the Ru-catalyzed BZ reaction, and it can be used as an external factor to suppress or restore oscillations. For the control of computing networks discussed below, I considered ϕ(t) in the form:(4)$$\phi \left(t\right)=W\cdot (1.0+\eta +\tanh\left(-\xi \cdot \left(t-t_{illum}\right)\right)$$
Such a definition of ϕ(t) involves a few parameters: *W* and η determine the limiting values of illumination at t→±∞. The value of illumination time $t_{illum}$ defines the moment of time when the most rapid changes in illumination occur, and ξ represents the rate at which the transition occurs. The minus sign in the argument of the tanh() function implies the transition from steady state (high illumination) towards oscillations (low illumination). In the presented simulations, I used fixed values: W=0.1, η=0.001 and ξ=10. For such parameters, the value of ϕ(t) is high (∼0.2) in the time interval $\left[0,t_{illum}-\delta \right]$ (δ=0.1) (cf. Figure 2a). Oregonator models with parameters given above predict stable steady state. For long times ($t>t_{illum}+\delta$), the value of $\phi_{j}\left(t\right)$ approaches 0.0001, which corresponds to oscillations.

The use of illumination time (or, in general, the inhibition time for oscillations inside a node) $t_{illum}$ to control oscillators is inspired by experiments in which oscillations of individual BZ droplets were regulated by blue light [54]. In these experiments, two illumination levels were used: a low one, for which the droplet was oscillating; and a high one, for which oscillations were inhibited. The transitions between steady state and oscillations predicted by the two-variable Oregonator model were in qualitative agreement with experiments.

### 2.3. The Model of a Network

An example illustrating the idea of a considered computing oscillator network is shown in Figure 3. The network is formed by three coupled chemical oscillators (nodes) graphically represented by circles. The nodes have different functions in networks. The upper one (#1) is a normal node whose illumination time is fixed. Two bottom nodes (#2) and (#3) are inputs of *x* and *y* coordinates, respectively. The arrows interlinking the oscillators represent reactions that exchange the activators between nodes. The arrows directed away mark the activator decay.

If the *j*th oscillator is a normal one, then the value of $t_{illum}\left(j\right)$ that appears in the definition of ϕ(t) (Equation (Equation 4)) does not depend on input values. Normal nodes are supposed to moderate the interactions within the network. The set of their illuminations can be regarded as a program executed by the network. If an oscillator is considered the input of a predictor $p^{i}$. then the value of $t_{illum}\left(j\right)$ is functionally related to $p^{i}$. For the analysis presented below, it is assumed that this function is defined by two parameters, $t_{start}$ and $t_{end}$, and has the form:(5)$$t_{illum}\left(j\right)=t_{start}+\left(t_{end}-t_{start}\right)\cdot p^{i}$$

Keeping in mind the symmetry of the Japanese flag problem, the values of $t_{start}$ and $t_{end}$ are the same for all predictors corresponding to *x*- and *y*-coordinates. This means that for record $\left(x_{n},y_{n},s_{n}\right)$:(6)$$t_{illum}\left(2\right)=t_{start}+\left(t_{end}-t_{start}\right)\cdot x_{n}$$
and
(7)$$t_{illum}\left(3\right)=t_{start}+\left(t_{end}-t_{start}\right)\cdot y_{n}$$

In a general case the values of $t_{start}$ and $t_{end}$ can be different for different predictors. The above form of relationship between the predictor value and illumination is simple. More complex functions can be used in specific cases and produce a network with higher accuracy. However, if the relationship between the input and the illumination time is represented by a complex function, then one can argue that part of the computation is done, not by the network, but by the specific form of this relationship.

Coupling between oscillators, indicated by arrows in Figure 3, is achieved by reactions that extend the original Oregonator model. I assume that the coupling is of the activatory type and occurs via the exchange of reactor activators between oscillators [49]. Let $U_{i}$ denote the activator of the *i*th node. The interactions between the nodes #*k* and #*j* appear as the results of reactions involving the activators $U_{k}$ and $U_{j}$: (8)$$\begin{array}{c} U_{j}+B_{j,k}\to U_{k}+C_{k} \end{array}$$(9)$$\begin{array}{c} U_{k}+B_{k,j}\to U_{j}+C_{j} \end{array}$$
with the reaction rate constants $k_{B,j,k}$ and $k_{B,k,j}$.

The changes in concentrations of $U_{k}$ and $U_{j}$ as the result of reactions (8) and (9) are described by:(10)$$\begin{array}{cc} \frac{du_{j}}{dt} & =-k_{B,j,k}b_{j,k}u_{j} \end{array}$$(11)$$\begin{array}{cc} \frac{du_{k}}{dt} & =-k_{B,k,j}b_{k,j}u_{k} \end{array}$$
and the changes in concentration of $U_{j}$ as the result of reaction (Equation 3) by:(12)$$\frac{du_{j}}{dt}=-k_{D}d_{j}u_{j}$$
In Equations (10)–(12), $b_{j,k}$, $b_{k,j}$ and $d_{j}$ denote concentrations of $B_{j,k}$, $B_{k,j}$ and $D_{j}$, respectively. We assume that these concentrations are high with respect to the concentrations of the activators involved, are the same for all oscillators, and remain almost constant during the network evolution. Therefore, they are not included in the model of network evolution. Let us introduce symbols $\alpha_{j}$ and $\beta_{j,k}$, defined as: $\alpha_{j}=k_{D}d_{j}$ and $\beta_{j,k}=k_{B,j,k}b_{j,k}$. Keeping in mind that values of $\alpha_{j}$ and $\beta_{j,k}$ can be modified by concentrations of $D_{j}$ and $B_{j,k}$, we can treat them as free parameters that can be easily adjusted. The values of $\alpha_{j}$ and $\beta_{j,k}$ are included in the optimization procedure. Let us also notice that all information on the network geometry is included in the values of $\beta_{j,k}$, because for non-coupled nodes $\beta_{j,k}=\beta_{k,j}=0$.

On the basis of the above assumptions, we can write equations describing the time evolution of the network:(13)$$\begin{array}{ccc} \frac{du_{j}}{dt} & = & \frac{1}{\varepsilon }\left(u_{j}-u_{j}^{2}-\left(fv_{j}+\phi_{j}\left(t\right)\right)\frac{u_{j}-q}{u_{j}+q}\right)-\left(\alpha_{j}+\sum_{i=1,m}\beta_{j,i}\right)u_{j}+\sum_{i=1,m}\beta_{i,j}u_{i} \end{array}$$(14)$$\begin{array}{ccc} \frac{dv_{j}}{dt} & = & u_{j}-v_{j} \end{array}$$
where *i*, *j* represent the *i*th and *j*th oscillator, respectively, and *m* is the number of oscillators in the network. This set of equations is solved numerically with a fourth-order RK algorithm [55] and time step of dt=0.0005.

I assume that the output information can be extracted from observation of the network evolution during the time interval $Z=[0,t_{max}]$. This assumption is essential. In many previous studies on chemical computing, it was assumed that the system reaches a specific steady-state that represents the answer. In the present approach, the output information can be read from the time evolution of a selected oscillator observed in a finite time interval, and what happens later is irrelevant to computation.

Let us notice that Equation (Equation 5) has a physical meaning for any value of $t_{illum}\left(j\right)$. If $t_{illum}\left(i\right)<0$, then $\phi_{i}\left(t\right)$ is small and the oscillator #i is active during the whole observation interval. When $t_{illum}\left(k\right)>t_{max}$, then $\phi_{k}\left(t\right)$ is large and the oscillator #k is inhibited within *Z* and does not oscillate.

In previous papers on classification with networks of interacting chemical oscillators, the number of activator maxima observed within the time interval $Z-\{0,t_{max}\}$ on a selected node (say #*j* ) was considered as the output [47,49]. However, there is a question: What is the maximum of activator? I assume that $u_{j}\left(t\right)$ should have a maximum in a strict mathematical sense. This means that if $u_{j}\left(t\right)$ has a maximum at $t_{0}$, then there exists ν>0 such that $u_{j}\left(t\right)<u_{j}\left(t_{0}\right)$ for all $t\in [t_{0}-\nu ,t_{0}+\nu ]\subset Z$. Therefore, if $u_{j}\left(t\right)$ is growing at the end of *Z*, then $u_{j}\left(t_{max}\right)$ is not regarded as a maximum. Moreover, the value of $u_{j}\left(t_{0}\right)$ should be larger than the activator concentration in the part of the oscillation cycle when the catalyst is in its reduced form. Here, I assume that the threshold value is 0.05. To illustrate maxima counting, let us consider the network with parameters given in the first row of Table 1. Figure 4 and Figure 5 show the time evolution of activator (red curves) and inhibitor (blue curves) at all nodes for two selected points of the flag. The green line marks the threshold for the activator maximum (0.05). The results in Figure 4 illustrate the time evolution of activator and inhibitor on all nodes when the coordinates of the input point are: (−0.25,0.25). The point characterized by such coordinates is located in the red area of the flag. For this input, there are two maxima of the activator at all network nodes. It can also be seen that on all nodes, the concentration of the activator is larger than 0.05 at the final stage of the evolution. However, the function u(t) is still growing at the end of *Z*, so the third maximum of u(t) is not observed. Figure 5 illustrates the time evolution of activator and inhibition on all nodes if the coordinates of the input point are: (0.29,−0.29). This point is located in the white region of the flag. Now, we observe three maxima of the activator at nodes #1 and #3, and two maxima at node #2. Therefore, if we consider the number of maxima on a selected oscillator as the network answer, then by selecting nodes #1 or #3, one can distinguish which of two points {(−0.25,0.25),(0.29,−0.29)} is white and which is red. Such classification is not possible if node #2 is considered the output. Selection of the output node can be made on the basis of correct answers for the records included in the training dataset.

Figure 6 shows another example of classification with a number of activator maxima recorded on the selected node. Here, we consider the time evolution of the activator (the red curve) and the inhibitor (the blue curve) on node #1 of the network defined by the parameters listed in the second line of Table 1. Figure 6a illustrates the time evolution of concentrations for the point outside the red area (−0.25,0.28), and Figure 6b for the point inside it (−0.39,−0.43). In the first case, no maximum of activator concentration is observed for times in $Z-\{0,t_{max}\}$; in the second case, we have a single maximum. Therefore, by observing the number of maxima on node #1, one can distinguish a red point from a white point, and a binary answer (0 or 1) is given. A classifying network with such properties can be regarded as highly optimized because the answer is obtained within a time similar to a single period of oscillation.

Figure 7 and Figure 8 illustrate another method for extracting the output information from the network evolution. This was inspired by [48], in which the output of an information-processing chemical BZ oscillator was related to the concentration of a selected reagent integrated over a specific time interval. Here we use the concentration of activator u(t) (red, Figure 7) and the concentration of inhibitor v(t) (blue, Figure 8) integrated over the observation time $\left[0,t_{max}\right]$ as the network output. In these figures, the shaded areas below the function represent the integrals $J_{u}=\int_{0}^{t_{max}}u_{1}\left(t\right)dt$ and $J_{v}=\int_{0}^{t_{max}}v_{3}\left(t\right)dt$, respectively. The integral value *J* is a real number, whereas an integer output is expected. In order to get such an output, I applied the following transformation:(15)output=floor(40·J)
where *J* stands for $J_{u}$ or $J_{v}$. For the integrated concentration of the activator (Figure 7), we obtained $J_{u}=0.101$ for the considered point inside the red area and $J_{u}=0.073$ for the point outside it. Therefore, the network answers were 4 and 2 for these points, respectively. The method applied to the integral of the inhibitor (Figure 8) produced the outputs $J_{v}=0.203$ for the considered point inside the red area and $J_{u}=0.181$ for the point outside it. In this case, the network outputs were 8 and 7, respectively. Therefore, the integrals of the activator and inhibitor can also be used to classify points of different colors on the Japanese flag.

### 2.4. Top-Down Design of Computing Networks

To define an information-processing chemical oscillator network, we have to specify many parameters:-The observation time $t_{max}$;-All parameters for a model of chemical oscillations inside a node; for the Oregonator model, they are ε, *q* and *f*;-Parameters $t_{start}$ and $t_{end}$ that translate an input value into the illumination of an input oscillator (cf. Equation (Equation 5));-The rates for reactions responsible for interactions between oscillators ($\alpha_{j}$, $\beta_{j,i}$);-Location of input and normal oscillators;-Finally, the illumination times for all normal oscillators $t_{illum}\left(i\right)$.

Obviously, a network with randomly selected parameters has a small chance of working as a good classifier. All parameters listed above should be optimized for executing the required function. I do not know any algorithm that allows for a straightforward design of the optimum oscillator network for a given problem. Still, we can apply the idea of supervised training for parameter optimization. Training means that we need a teacher, and in our optimization, it is a specific database $T_{A}$ that contains a sufficient number of records related to the considered problem *A* [56].

The network should include the output oscillator whose time evolution is transformed into the network answer. Here I consider two types of answers. One is the number of activator maxima observed in the time interval $\left[0,t_{max}\right]$. Another is the integral of the activator or inhibitor concentration. If the network parameters are known, the output oscillator can be located.

In order to find which oscillator should be used as the output, one can calculate the mutual information $I(S;O_{j})$[57] between the discrete random variable *S* of record types in the training dataset $T_{A}$ ($S=\{s_{n},n=1,N\}$) and the discrete random variable $O_{j}$ representing the output of the *j*th oscillator in the network when the predictors of *n*th database record are used as the network input ($O_{j}=\left\{o_{j}\left(n\right),n=1,N\right\}$). The mutual information $I(S;O_{j})$ can be calculated as:(16)$$I\left(S;O_{j}\right)=H\left(S\right)+H\left(O_{j}\right)-H\left(S,O_{j}\right)$$
where H() is the Shannon information entropy [58], and the discrete random variable $\left(S,O_{j}\right)=\left\{\left(s_{n},o_{j}\left(n\right)\right),n=1,N\right\}$. The oscillator #i for which the mutual information between *S* and $O_{i}$ is maximal is used as the network output. The mutual information calculated for the output oscillator is considered the measure of network fitness:(17)$$Fitness=max_{j\in \{1,2,3\}}\;I(S;O_{j})$$
It can be expected that in the majority of cases, optimization based on mutual information leads to a classifier with the highest accuracy [59].

The fitness function based on mutual information does not require specifying how the network evolution translates into the network output. However, if we know how to link the output $o_{j}\left(n\right)$ with the record type, then we can calculate the accuracy on node #*j* for training dataset $E_{j}$ as the ratio between correctly classified cases and all cases of $T_{A}$. For example, we can relate a specific number of activator maxima to the color of a point based on the majority of cases obtained for the training dataset. Therefore, we can alternatively define Fitness as:(18)$$Fitness=max_{j\in \{1,2,3\}}\;E_{j}$$
Both formulae (Equation 17) and (Equation 18) allow us to locate the output oscillator in the network.

It has been demonstrated that evolutionary optimization [60] oriented towards obtaining the best classifier for a representative training dataset of the problem can lead to a computing network that performs the anticipated task with reasonable accuracy [45,46,49]. In this approach, we represent the parameters as a code that undergoes recombination between the fittest members and mutations of an offspring. For recombination, two networks are selected, and their parameters are randomly separated into two parts. Next, an offspring is generated by combining one part of the first network with the other part of the second one. At this step, the function of an oscillator (input, normal) and illumination times of normal oscillators are copied to the offspring. For the next step, mutation of all parameters of the newborn offspring is considered. The probability of mutation rate is 0.5 per step, and the change in parameter value does not exceed 20%. Each generation of networks consists of the same number of elements. It includes a few fittest networks from the previous generation that are copied without changes in parameters. The remaining members of the next generation are offspring created by recombination and mutation operations applied to oscillators from the top 50% of networks from the previous population.

However, optimization of all parameters as mentioned above represents a computational problem of very high complexity. Before starting optimization, we introduce a number of simplifications:(1)My attention is restricted to classifiers formed by m=3 oscillators;(2)There have to be input oscillators for each coordinate in the network and a normal oscillator. Keeping in mind the symmetry of the considered network, we can assume that node #1 is the normal oscillator and nodes #2 and #3 are the inputs of *x*- and *y*-coordinates, respectively;(3)The system symmetry reduces the number of parameters in the networks because: $\alpha_{2}=\alpha_{3}$, $\beta_{1,2}=\beta_{1,3}$, $\beta_{2,1}=\beta_{3,1}$ and $\beta_{2,3}=\beta_{3,2}$.

After all these simplifications, the network is fully defined by the parameters listed in Table 1.

## 3. What Is the Color of a Point on the Japanese Flag? (As Seen by the Networks)

Here I present the results of network optimization. The training dataset is composed of 1000 records; 501 represent points inside the sun area and 499 are points outside it. The location of points can be seen, for example, in Figure 9. The applied evolutionary algorithm is a standard one, and it has been described in [45,46,49]. Optimization starts with 80 networks with randomly selected parameters. All networks in the initial population are evolved for maximum fitness. The fittest 10% (8) of the networks are copied to the next generation. The population of the next generation is completed to 80 networks by classifiers obtained by recombination and mutation operations applied to networks randomly selected from the upper 50% of the fittest. Next, the evolution step is repeated on the new population. The optimization procedure is executed for 1000 steps.

Figure 9, Figure 10, Figure 11 and Figure 12 illustrate the answer of the fittest networks obtained for two Oregonator models and three different methods of interpreting the network output. The parameters of the discussed networks are given in Table 1. In all cases, the network output is based on the time evolution of a selected node in the network. For the first and second networks (Figure 9 and Figure 10), this is the number of activator maxima observed on a selected node in the time interval $\left[0,t_{max}\right]$. For the third and fourth network, this is the integrated concentration of activator on a selected node (Figure 11) and the integrated concentration of inhibitor on a selected node (Figure 12). In all networks, the classification rule is derived after network optimization.

Figure 9 shows the probability distribution of the number of activator maxima for the network defined by the parameters listed in the first line of Table 1. The red and blue bars correspond to points located in the sun and outside it, respectively. On nodes #2 and #3, we observe a large number of records representing points both in and out of the sun area that produced two maxima of the activator (cf. Figure 9c,d). On the other hand, for node #1, the number of activator maxima is two for most of the records representing points inside the sun area, and three maxima are observed for a large majority of points outside it. Therefore, node #1 is considered the output, and the classification rule is:-If one or two maxima are observed, then the record represents a point in the sun area of the training dataset;-If three or more maxima are observed, then the record represents a point outside the sun area.

Such a rule incorrectly classifies 24 points located outside the sun and 8 points located inside the sun area. Therefore, the accuracy of this network with this classification rule on the training dataset is 96.8%. The geometry of correctly and incorrectly classified records of the training database is illustrated in Figure 9b. In this figure, as well as on similar figures for the other considered classifiers, the red dots represent points inside the sun area that are correctly recognized by the network. Similarly, the yellow dots represent points outside the sun area that are correctly recognized by the network. The green dots are points located within the sun area, but the network classifies them as belonging to the white region. Finally, the blue dots are misclassified points located in the white region. In order to see the errors more clearly, the dots corresponding to incorrectly classified points are larger than those representing correct answers. The location of errors at the edge of the red area is non-homogeneous and suggests the existence of regions where errors dominate.

Similar results, but for the network defined by the parameters listed in the second line of Table 1, are shown in Figure 10. Again, node #1 is the network output because on nodes #2 and #3, we observe one activator maximum for a large number of records representing points located both in and out of the sun area (cf. Figure 10c,d). For node #1 (Figure 10a), most records representing points inside the sun area produced a single maximum, and for most points located outside the sun, no maximum was observed (cf. Figure 6b). Therefore, node #1 is the output, and the classification rule is:-If a single maximum of the activator is observed, then the record represents a point in the sun area of the training dataset;-If we record no maxima, then the processed data represent a point outside the sun area.

Such a rule incorrectly classifies just three points located outside the sun and seven points located inside the sun area. The accuracy of the second considered network with the classification rule listed above is 99% on the training dataset. The geometry of correctly and incorrectly classified records of the training database is illustrated in Figure 10b. All incorrectly classified points are located close to the boundary between the sun and the white region. Let us notice that the accuracy is 2% higher than for the previous network, and the only difference between these networks is the character of oscillations (cf. Figure 2b,c). This result suggests that “softer” oscillations may give networks with higher accuracy. This is the reason why the Oregonator Model II was used to investigate the alternative methods for relating the time evolution of concentration on a selected node with the network output.

Figure 11 presents results for the network defined by the parameters listed in the third line of Table 1. The network output is defined as: $J_{u}=\int_{0}^{t_{max}}u_{j}\left(t\right)dt$, where *j* denotes the output node. The range of $J_{u}$ values is divided into subintervals $I_{k}$ with length 0.025 as follows: $I_{k}=\left[k*0.025,\left(k+1\right)*0.025\right)$ for k∈1,2,3,4. Figure 11a illustrates the probability distribution of $J_{u}$ values for output node #1. The classification rule can be formulated as follows:-If the value of $J_{u}\geq 0.1$, then the record represents a point in the sun area of the training dataset;-If the value of $J_{u}<0.1$, then the record represents a point outside the sun area.

Such a rule incorrectly classifies just 2 points located inside the sun area and 11 points located outside it. All of the misclassified points are located close to the sun edge (cf. Figure 11b). The accuracy of the third of the considered networks with the above classification rule is 98.7% on the training database.

Keeping in mind the problem symmetry, we expect that node #1 should be the output. This was confirmed in the first three considered networks. However, node #1 gives poor accuracy for the network defined by the parameters listed in the fourth line of Table 1. For this network, the output is defined as: $J_{v}=\int_{0}^{t_{max}}v_{j}\left(t\right)dt$, where *j* denotes the output node. As for the previously discussed network, the range of $J_{v}$ values is divided into subintervals $I_{k}$. Figure 12a illustrates the probability distribution of $J_{v}$ values for output node #1. The records representing points in the sun area and those located outside it return values of $J_{v}$ in the third subinterval $I_{3}$. Figure 12b shows the distribution of correctly and incorrectly classified records. For the majority rule derived from Figure 12a, all points located outside the sun are correctly classified, but there are many points inside the sun that are classified as located outside. Figure 12c illustrates the probability distribution of $J_{v}$ values for output node #3. We observe a nice separation of records representing points in the sun area from those located outside. On the basis of this result, we formulate the following classification rule:-If the value of $J_{v}\geq 0.2$, then the record represents a point in the sun area of the training dataset;-If the value of $J_{v}<0.2$, then the record represents a point outside the sun area.

Such a rule incorrectly classifies just five points located inside the sun area and nine points located outside it. All of the misclassified points are located close to the sun’s edge (cf. Figure 12d). The accuracy of the fourth considered network with the above classification rule is 98.6% on the training database. Network symmetry is reflected by high classification accuracy based on the inhibitor’s time evolution on node #2. If we use this node as the output, we get classification accuracy of 97.9%. It can be expected that the difference between the accuracy with node #2 versus node #3 as the output is related to the choice of the training dataset, and in a perfectly balanced choice, both of these nodes should have the same accuracy.

## 4. Discussion and Conclusions

In the previous section, I demonstrated that three-node networks can be optimized to determine the color of a point in the Japanese flag with high accuracy. However, the accuracy was achieved on the training dataset, raising the question of whether the results shown in Figure 9, Figure 10, Figure 11 and Figure 12 are general, or if they reflect correlations inevitable in the small set of randomly selected points that formed the training database.

To produce a stronger argument for the accuracy of optimized networks, I verify them on a large test dataset that contained 100,000 records. The points are generated randomly, and the test dataset includes 49,916 records corresponding to points inside the sun area and 50,084 points outside it.

The answers of the networks defined by the parameters listed in Table 1 to the records of the testing dataset are presented in Figure 13. Red and yellow dots represent correctly classified points located inside and outside the red area of the flag. To reduce the figure size, the number of points is limited to 10,000. The green color marks points located in the red area that are wrongly classified as being located in the white part. The blue color denotes points from outside that are classified as belonging to the sun area. In all cases, the classification is accurate, and the flag is nicely represented.

Figure 13a shows the results for the networks defined by the parameters listed in the first line of Table 1. As suggested by the results in Figure 9, the number of activator maxima on node #1 is used as the output. The majority rule is:-If one or two maxima are observed, then the point is within the sun area;-If three or more maxima are observed, then the point is outside the sun area.

Such a rule incorrectly classifies 3216 points located outside the sun (blue in Figure 13a) and 1111 points located inside the sun area (green in Figure 13a). Therefore, the accuracy of this classifier is 95.668%. The sun area considered in this paper is more symmetrically oriented in the coordinate system than the Japanese flag located in the unit square [0,1]×[0,1] studied in [47]. For such a flag location the optimized classifier is characterized by: $t_{max}=20.23$, $t_{start}=3.78$, $t_{end}=12.10$ and $t_{illum}\left(3\right)=6.37$. Its accuracy tested on 100,000 records is 95.145%. However, to obtain this result, it was assumed that the values of parameter $\alpha_{j}$ were identical for all nodes (=0.849). Moreover, the values of $\beta_{i,j}=0.251$ were the same for all pairs of nodes. These assumptions definitely decreased the accuracy of the network reported in [47] if compared to the one defined in the first row of Table 1. Therefore, I expect that the accuracy of the three-oscillator classifier that codes the output in the number of activator maxima on a selected node does not systematically depend on the location of the flag in the coordinate system. The points classified as the sun (red and blue Figure 13a) form a characteristic horned disk already observed for a three-node classifier optimized to recognize the Japanese flag in the unit square [0,1]×[0,1] [47]. The fact that the horned shape is repeated by optimization using a training database containing points with coordinates in different ranges suggests that this shape is related to the parameters of the Oregonator Model I.

The horned shape is not observed if one still uses the number of activator maxima as the output but considers another set of Oregonator parameters (Model II, (cf. Figure 13b)). A trace of undeveloped horns can be seen on the decreasing diagonal, but the sun disk is well represented. The classification rule (cf. Figure 10a) incorrectly predicts the color for 446 points located outside the sun (blue in Figure 13b) and 1192 points located inside the sun area (green in Figure 13b). The classifier accuracy is 98.362%. This result shows that the change in the oscillator model can improve the accuracy and modify the geometry of incorrectly classified points.

Figure 13c,d represent classifiers where the integrals of activator and inhibitor observed on the output node were used as the output. The analysis of output related to activator concentration is shown in Figure 13c. The points inside the sun were those for which $J_{u}\geq 0.1$. This rule incorrectly predicts the color for 2021 points located outside the sun (blue in Figure 13c) and only 401 points located inside the sun area (green in Figure 13c). The corresponding accuracy is 97.578%.

In the case of output related to inhibitor concentration (Figure 12c,d), the points inside the sun were those for which $J_{v}\geq 0.2$. This rule incorrectly predicts the color for 1409 points located outside the sun (blue in Figure 13d) and only 691 points located inside the sun area (green in Figure 13d). The corresponding accuracy is 97.900%. The use of node #3 as the output is suggested due to its having the highest accuracy obtained on the training dataset. However, slightly higher accuracy (97.963%) is obtained if node #2 acts as the output. This indicates that even for such a simple problem as the Japanese flag one, the size of the training dataset is important to obtain the correct rule of classification.

The comparison between the accuracy of described classifiers shows that the method based on the integration of activator or inhibitor concentration is as good as the output represented by the number of the corresponding maxima. In the method of maxima counting, there was a magic parameter—the threshold value for acceptance of maximum. The method based on integration does not need to involve such additional parameters to obtain the result and seems to be a promising candidate for future investigation on the computing potential of networks composed of interacting chemical oscillators.

## Figures and Tables

**Figure 1 entropy-24-01054-f001:**
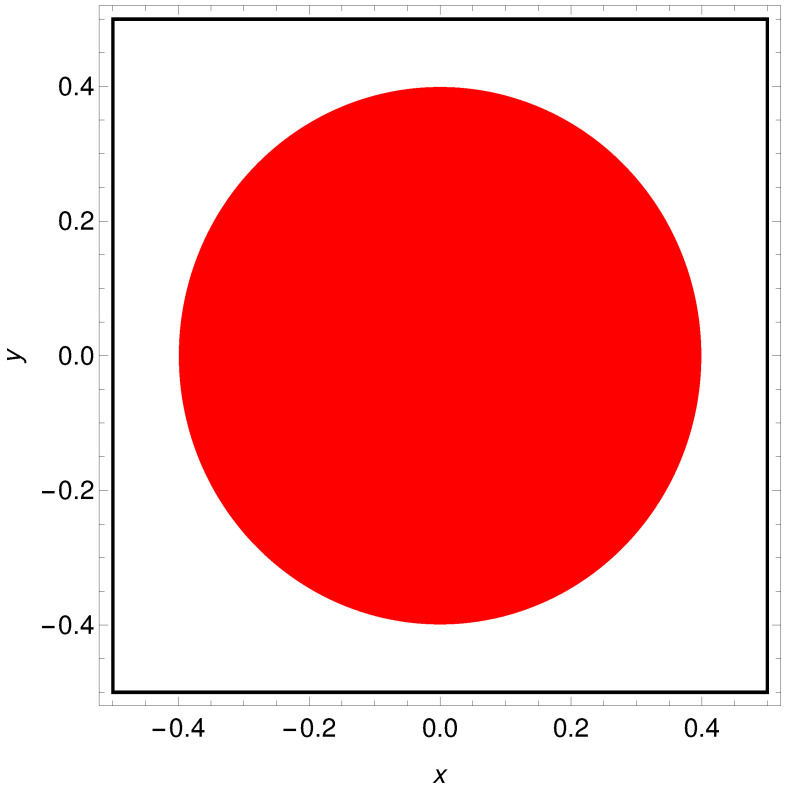
The geometrically inspired problem of determining the color of a randomly selected point located on the Japanese flag formed by the central red disk and the surrounding white area. The flag is represented by the Cartesian product [−0.5,0.5]×[−0.5,0.5]), and the disk radius is $r=1/\sqrt{(}2\pi )$; thus, the areas of the sun and the white region are equal.

**Figure 2 entropy-24-01054-f002:**
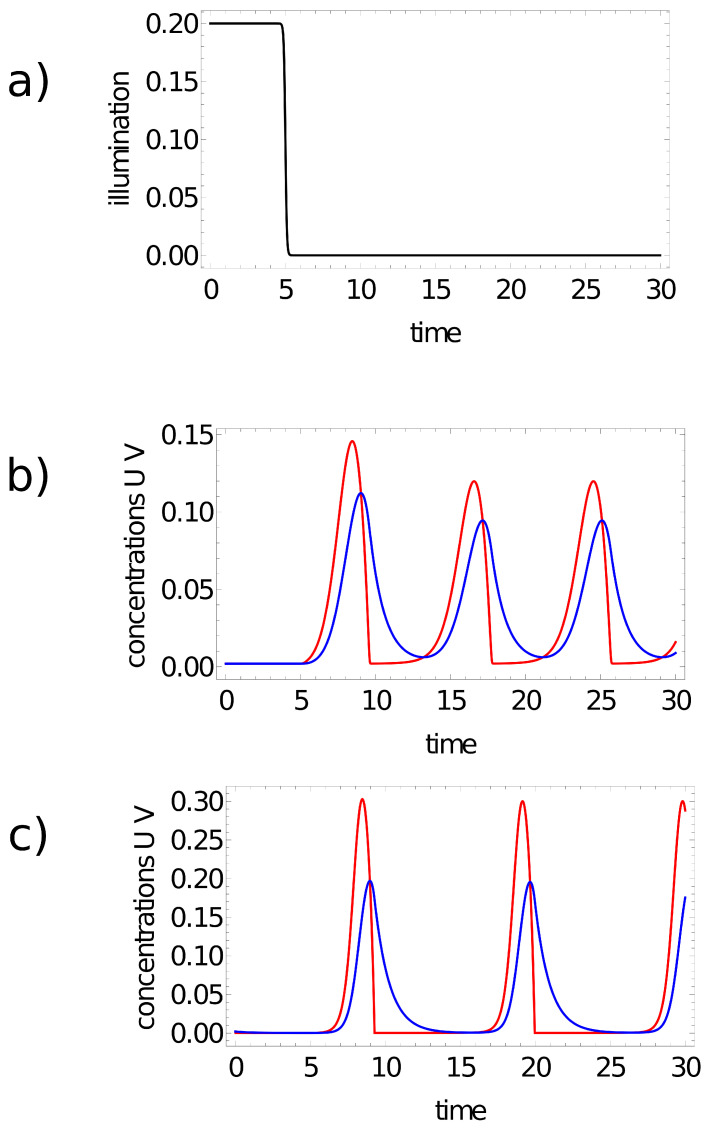
(**a**) Time-dependent illumination $\phi \left(t\right)=(1.001+\tanh\left(-10*\left(t-t_{illum}\right)\right))/10$ for $t_{illum}=5$. (**b**,**c**) The character of oscillations for the 2-variable Oregonator models used in simulations: (**b**) Model II: f=1.1, q=0.002, ϵ=0.3; (**c**) Model I: f=1.1, q=0.0002, ϵ=0.2. Red and blue curves represent concentrations of activator (u) and inhibitor (v), respectively. The values of α are 0.5 (**b**) and 0.7 (**c**).

**Figure 3 entropy-24-01054-f003:**
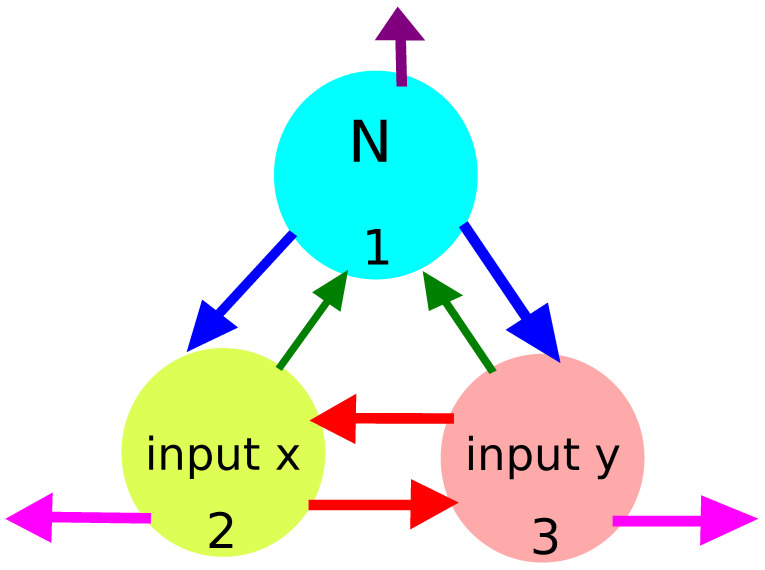
The idea of a computing oscillator network. Circles represent network nodes that are chemical oscillators. The nodes have different characteristics. The upper one (#1) is a normal one, and its illumination function is fixed. The bottom nodes (#2) and (#3) are inputs of *x* and *y* coordinates, respectively. The arrows interlinking oscillators represent reactions that exchange the activators between nodes. The arrows directed away mark activator decay (reaction 3).

**Figure 4 entropy-24-01054-f004:**
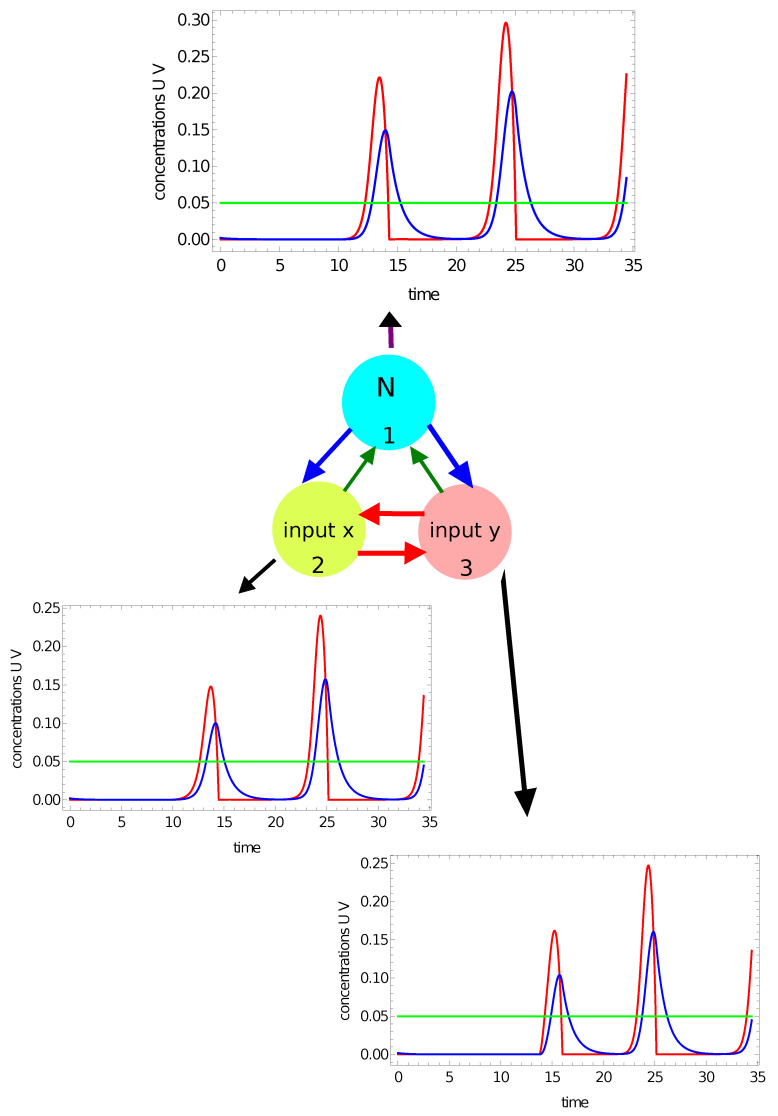
The time evolution of activator (the red curves) and inhibitor (the blue curves) observed on all nodes of the network defined by the parameters listed in the first line of Table 1. The coordinates of the input point are: (−0.25,0.25). The green line marks the threshold for the activator maximum. There are 2 maxima at all nodes in the network.

**Figure 5 entropy-24-01054-f005:**
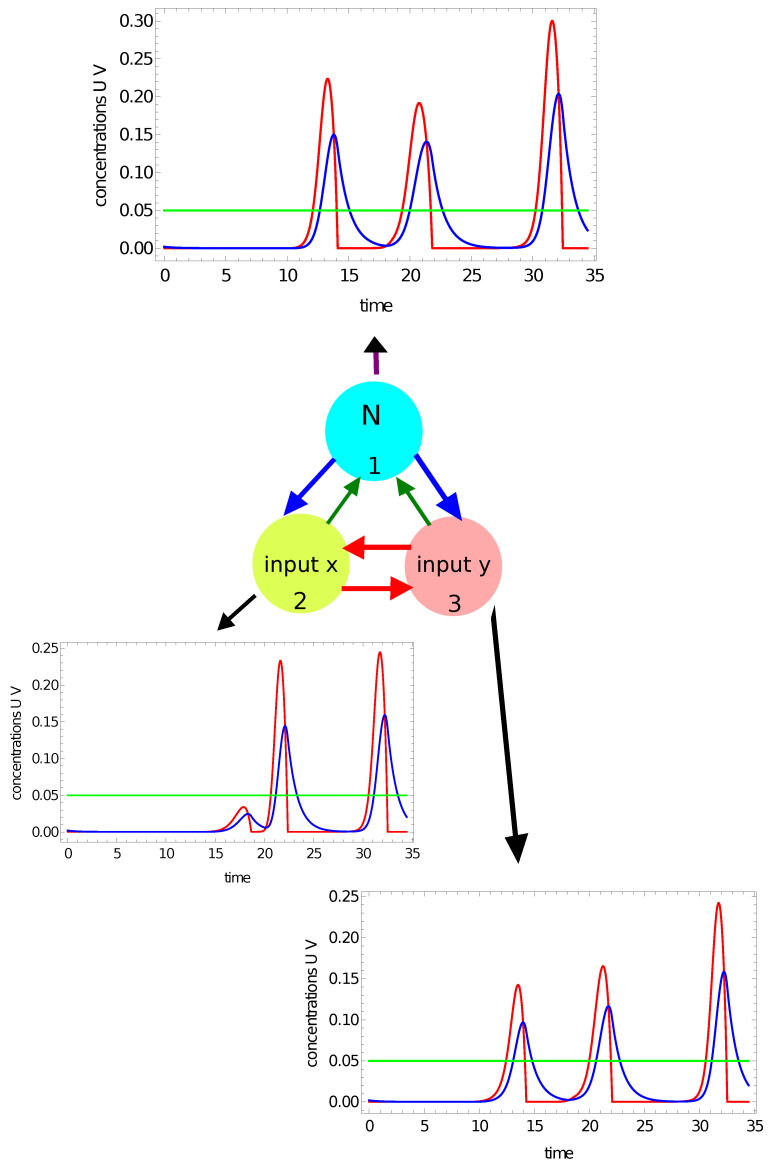
The time evolution of activator (the red curves) and inhibitor (the blue curves) observed on all nodes of the network defined by the parameters listed in the first line of Table 1. The coordinates of the input point are: (−0.29,0.29). The green line marks the threshold for the activator maximum. There are 3 maxima of u(t) on nodes #1 and #3 and two maxima of u(t) on node #2 within the observation time $\left[0,t_{max}\right]$. The green line marks the threshold for the activator maximum.

**Figure 6 entropy-24-01054-f006:**
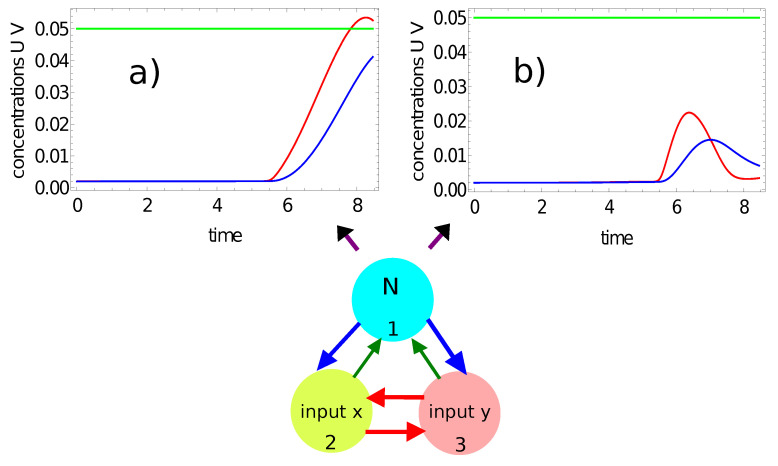
The time evolution of the activator (the red curve) and the inhibitor (the blue curve) on node #1 of the network defined by the parameters listed in the second line of Table 1: (**a**) $u_{1}\left(t\right)$ and $v_{1}\left(t\right)$ for the point inside the red area (−0.25,0.28); (**b**) $u_{1}\left(t\right)$ and $v_{1}\left(t\right)$ for the point outside the red area (−0.39,−0.43).

**Figure 7 entropy-24-01054-f007:**
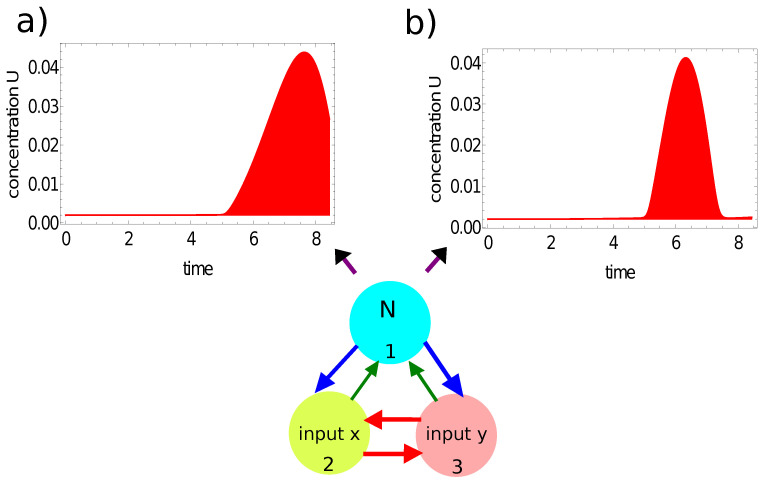
The time evolution of the activator at node #1 of the network defined by the parameters listed in the third line of Table 1: (**a**) $u_{1}\left(t\right)$ for the point (−0.25,0.28) located inside the red area; (**b**) $u_{1}\left(t\right)$ for the point (−0.39,−0.43) located outside the red area. The red shaded area below the function represents the integral of $J_{u}=\int_{0}^{t_{max}}u_{1}\left(t\right)dt$, considered as the network output.

**Figure 8 entropy-24-01054-f008:**
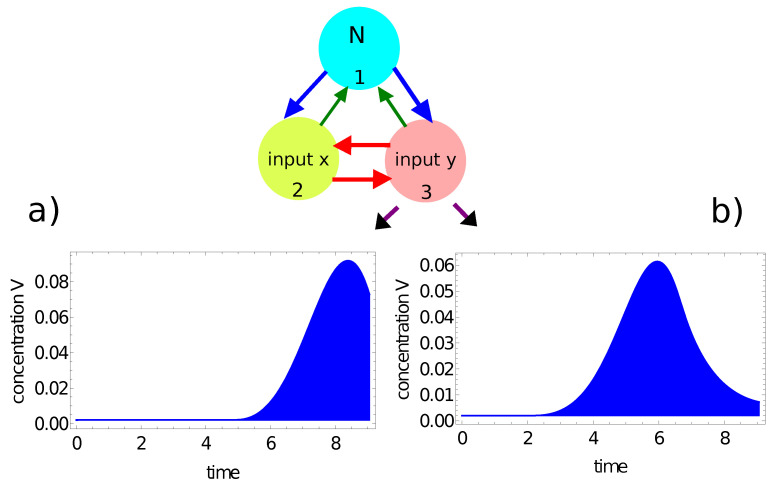
The time evolution of the inhibitor at node #3 of the network defined by the parameters listed in the fourth line of Table 1: (**a**) $v_{3}\left(t\right)$ for the point (−0.25,0.28) located inside the red area; (**b**) $v_{3}\left(t\right)$ for the point (−0.39,−0.43) located outside the red area. The blue shaded area below the function represents the integral $J_{v}=\int_{0}^{t_{max}}v_{3}\left(t\right)dt$, considered as the network output.

**Figure 9 entropy-24-01054-f009:**
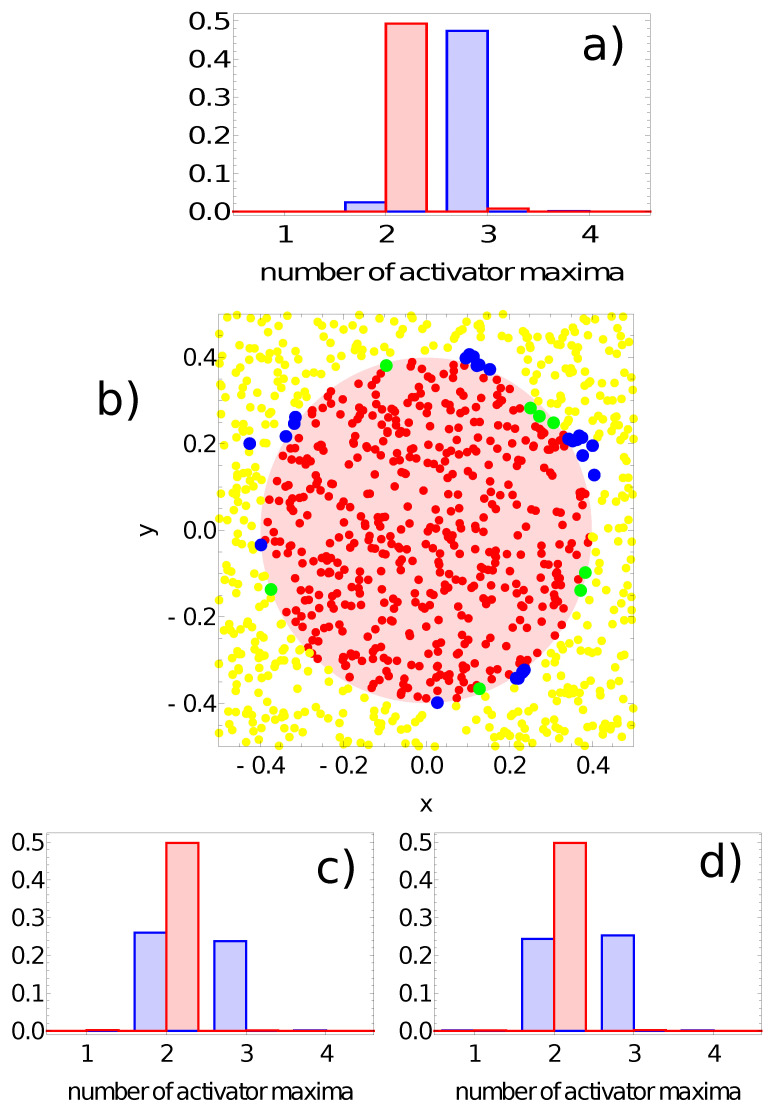
The answer of the network defined by the parameters listed in the first line of Table 1 to the records of the training dataset. Subfigures (**a**,**c**,**d**) are probability distributions of obtaining a given number of activator maxima on nodes #1, #2 and #3, respectively. The red bars correspond to points inside the red area; the blue bars refer to points outside the red area. Subfigure (**b**) illustrates correctly (yellow and red) and incorrectly (green and blue) classified points of the training dataset when node #1 is used as the output.

**Figure 10 entropy-24-01054-f010:**
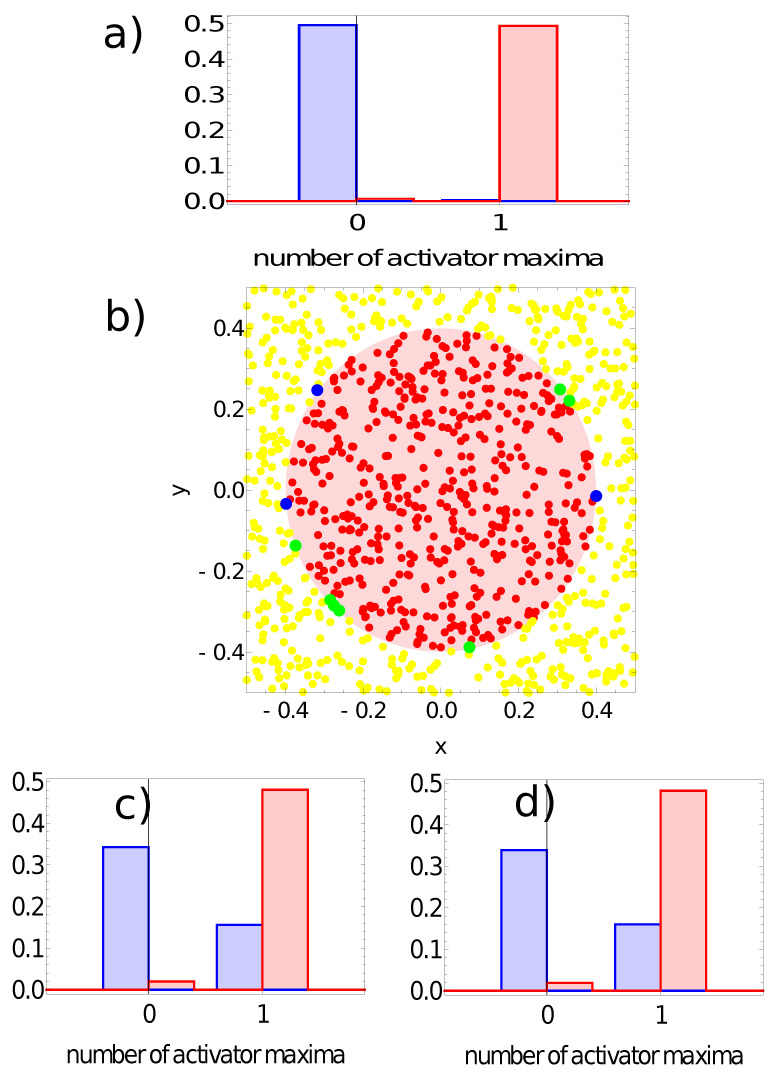
The answer of the network defined by the parameters listed in the second line of Table 1 to the records of the training dataset. Subfigures (**a**,**c**,**d**) are probability distributions of obtaining a given number of activator maxima on nodes #1, #2 and #3, respectively. The red bars correspond to points inside the red area; the blue bars refer to points outside the red area. Subfigure (**b**) illustrates correctly (yellow and red) and incorrectly (green and blue) classified points of the training dataset when node #1 is used as the output.

**Figure 11 entropy-24-01054-f011:**
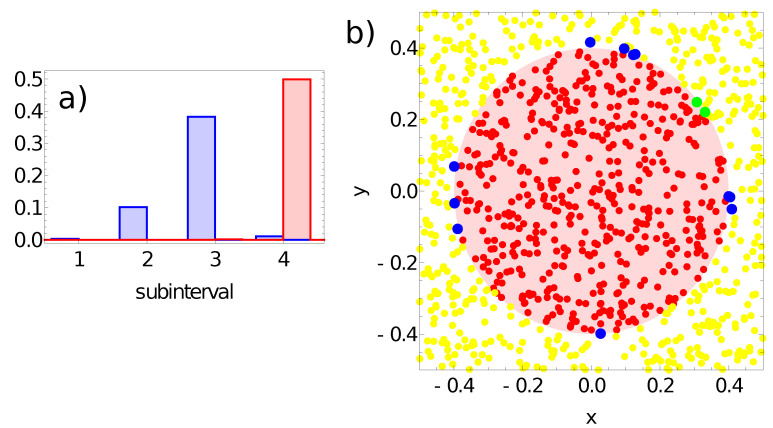
The answer of the network defined by the parameters listed in the third line of Table 1 to the records of training dataset. (**a**) The probability distribution of obtaining the value of $J_{u}=\int_{0}^{t_{max}}u_{1}\left(t\right)dt$ in the intervals [k∗0.025,(k+1)∗0.025) for k∈{1,2,3,4}. The red bars correspond to points inside the red area; the blue bars refer to points outside the red area. Subfigure (**b**) illustrates correctly (yellow and red) and incorrectly (green and blue) classified points of the training dataset.

**Figure 12 entropy-24-01054-f012:**
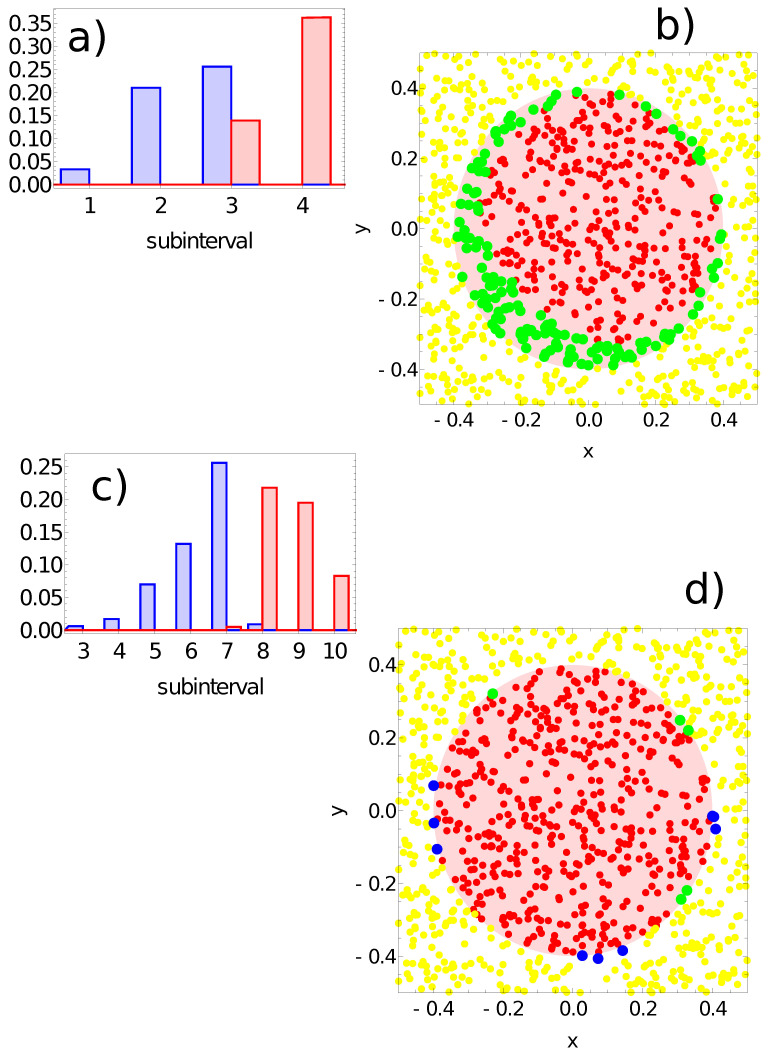
The answer of the network defined by the parameters listed in the fourth line of Table 1 to the records of training dataset. (**a**) The probability distribution of obtaining the value of $J_{v}=\int_{0}^{t_{max}}v_{1}\left(t\right)dt$ in the intervals [k∗0.025,(k+1)∗0.025) for k→{1,2,3,4}. The red bars correspond to points inside the red area; the blue bars refer to points outside the red area. Subfigure (**b**) illustrates correctly (yellow and red) and incorrectly (green) classified points of the training dataset. As in (**a**,**b**) but for node #3: (**c**) the probability distribution of obtaining the value of $J_{u}=\int_{0}^{t_{max}}v_{3}\left(t\right)dt$ in the intervals [k∗0.025,(k+1)∗0.025) for 3≤k≤10. Subfigure (**d**) illustrates correctly (yellow and red) and incorrectly (green and blue) classified points of the training dataset.

**Figure 13 entropy-24-01054-f013:**
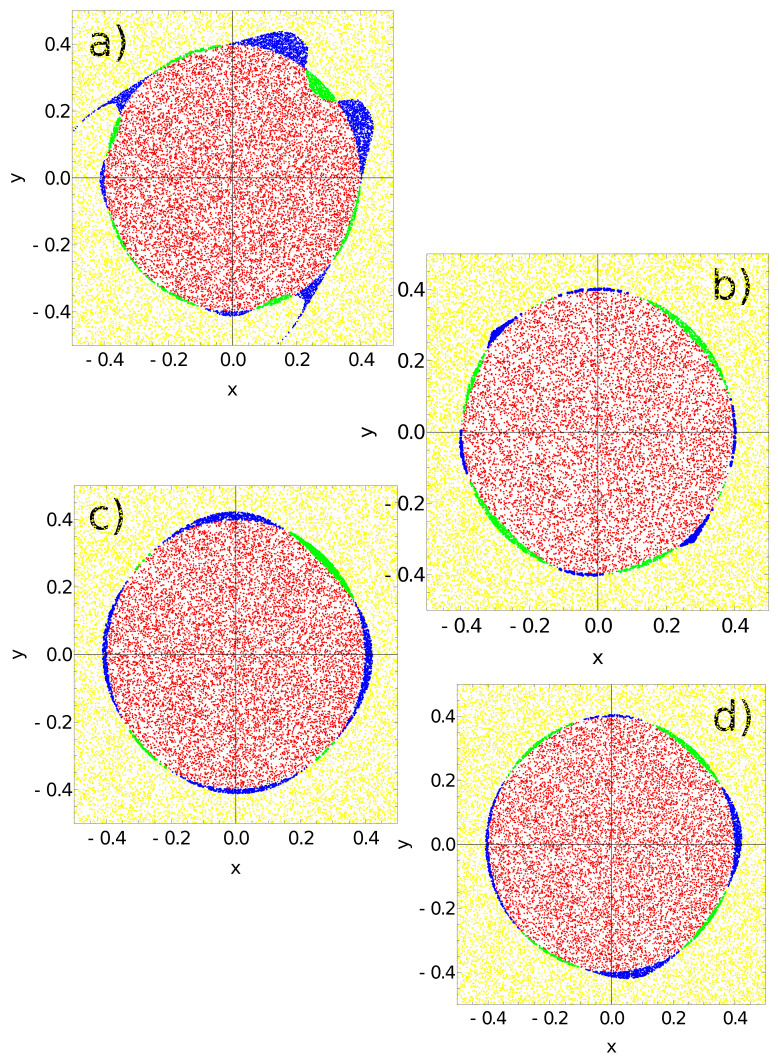
The answer of networks defined by the parameters listed in Table 1 to a testing dataset of 100,000 records. Yellow and red points are classified correctly. The network gives a wrong answer on points marked green (they belong to the sun but are classified as located outside it) and points marked blue (they are located outside the sun but are classified as belonging to the red area). Subfigures (**a**–**d**) correspond to networks with parameters listed in lines 1–4 of Table 1, respectively. The accuracy of these networks is (**a**) 0.957, (**b**) 0.984, (**c**) 0.976 and (**d**) 0.979, respectively.

**Table 1 entropy-24-01054-t001:** The parameters of networks that give the best correlations between the time evolution of the output oscillator and the point color.

Oregonator	Method	$t_{\max}$	$t_{\mathrm{start}}$	$t_{\mathrm{end}}$	$t_{\mathrm{illum}}\left(1\right)$	$\alpha_{1}$	$\alpha_{2}=\alpha_{3}$	$\beta_{1,2}=\beta_{1,3}$	$\beta_{2,1}=\beta_{3,1}$	$\beta_{2,3}=\beta_{3,2}$
Model I	activator maxima	34.4	11.6	19.9	10.1	0.87	0.72	0.16	0.43	0.29
Model II	activator maxima	8.45	3.77	8.03	5.42	0.96	0.46	0.53	0.38	0.42
Model II	u-integral	8.43	3.77	7.41	5.00	0.65	0.50	0.83	0.26	0.29
Model II	v-integral	9.06	3.77	7.56	5.71	0.75	0.44	0.60	0.29	0.33

## Data Availability

All presented data are available from the author.

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
