# Peer review of "Information Processing Using Networks of Chemical Oscillators"

_entropy, 2022, doi:10.3390/e24081054_

Round 1

Reviewer 1 Report

I have read with great interest the reviewed work, which is a type of review discussing an extremely interesting and rarely studied problem of the use of chemical dynamical systems for information processing. A significant part of this work is based on the latest work of the author's and coworkers, but also contains a lot of other information, showing the place of the present considerations against the background of the unstable chemical logic gates approach. This time, the procedure is based on the Belousov-Zhabotinsly photosensitive reaction model (modified two-dimensional Oregonator), from which networks analyzed using the evolutionary algorithm were created.

I consider this work to be a far from clichéd description.  It contains  quite novel (i.e. taken from original papers used for preparing this review) concepts and therefore I do support its publication. In addition, the inclusion in one synthetic study of the methodology of the procedure, described earlier in various publications, facilitates the understanding of the adopted procedure. Last but not least, a rich list of references allows the reader to have a broader understanding of the issues of the application of dynamic chemical systems in information processing.

In conclusion, I believe that this carefully prepared review should be published in its current form. My only minor suggestion is to add the words "A review" to the end of the title of the paper, to make the type of this work immediately clear.

Author Response

I am grateful for the report and positive comments. The words: A review have been added at the end of title.

Reviewer 2 Report

This article deals with the problems that can be solved with oscillator networks, a strategy often used in the study of a complex mixture of reacting chemical compounds. In this case, we speak of coupled chemical oscillators, and a typical example of an oscillating reaction is the Belousov-Zhabotinsky reaction. In this paper, the author applies this approach to a geometrically inspired problem involving the determination of the color of a randomly selected point on the Japanese flag. He discusses the input information, the way to read the network computation result, and describes the idea of top-down optimization of computational networks, which has an advantage over the bottom-up approach. In the analysis, it is shown that the top-down design of computational networks involves Shannon information entropy, a quantity whose physical interpretation has been puzzling for many years. However, it is demystified by accepting that it is only a statistical quantity. If a quantity is not a physical quantity, then there is no physical interpretation at all.

In my opinion, the results of this paper are of interest to the readers of Entropy and the manuscript is suitable for publication in that journal. I therefore recommend publication without further review.

Just a few typos:

In various places: Japan flag → Japanese flag
Line 188: Ont the other hand ...
Line 202: Preposition of is written twice.

Author Response

I am grateful for the positive opinion on the manuscript. The typos listed have been corrected. 

Reviewer 3 Report

This is a nice manuscript. Publication in "Entropy" is warranted. I will briefly summarize the contents and offer suggestions to improve the presentation.

The author of the manuscript already has a number articles behind his name on how coupled oscillating chemical reactors can operate like a "wet" neural-network computer. This manuscript is a good an original addition. The theory offers an elegant synthesis of different scientific disciplines. At the basis we have the well-known Belousov-Zhabotinsky reaction. The chemical kinetics of this reaction is described by Equations 2 and 3 in the manuscript. Coupling different reservoirs leads to Equations 12 and 13 of the manuscript. The reaction (and thus the oscillation) can be inhibited by shining light on the reservoir. The light intensity is the function \phi(t) in the aforementioned equations.  The function \phi_i(t) is the input for the i-th reactor. The \phi(t)'s is what makes the system non-autonomous. Characteristics of the chemical oscillations constitute the output of the computer. The author counts the number of maxima in a certain time interval, but also explores other options. 

As a practical example the "Japanese Flag Problem" is taken on. This flag is a red circle on a white surface. The idea is the encode the location (x,y) of a point on the flag in the illumination patterns \phi_i(t) and to let the oscillations tells you whether this is a red point or a white point.  After "training" the network and thereby evolving to the optimal parameter values, an accurate white-or-red predictor is developed.

There are a few points in the manuscript where I got puzzled and confused. There is ample room for improvement of the presentation. 

Page 8, lines 240-247, is the only clarification of how the position (x,y) on the flag is translated into illumination patterns \phi(t). This passage and the Equation 6 that precedes it are very cryptic. Earlier in the manuscript, in Section 2.1, the author describes the logic of classification in set-theoretical terms. This approach and Equation 1 may confuse and turn off some readers early in the manuscript. The q in Equation 1 is a different from the q in Equation 2 and this may also confuse. I am still wondering whether the p in Equation 6 is related to the p's in Equation 1. My recommendation is to forgo the formalism with Equation 1 and come up with a more solid and pragmatic explanation of how x and y translate into the inputs \phi_i(t). The mathematical-logic structures, moreover, have been explained in other published work of the author. 

Likewise forgone can be the information-theoretical explanation on lines 348-360 and containing Equations 15, 16, and 17. That passage is a bit of a digression. This material is also covered in other published works. The explanation is too brief to be a clear and informative addition. The results of the manuscript can be understood and appreciated without involving Shannon entropy and mutual information.

Let me finally point out a number of mishaps, typographical and other. 

1) line 22 "live in"

2) line 36 "many reports chemical realization"?

3) The custom is to once write "Belousov-Zhabotinsky (BZ)" and next consistently use only "BZ". 

4) Reference 39 is cited before reference 38

5) line 159 "can is"?

6) After Equations 2 and 3, there is talk about an "event-based model" that others studied. I don't know what such a model is. Many readers will not know. Lines 144-147 are puzzling. Get rid of it. Rewrite the explanation of Equations 2 and 3 to make it clearer. Say that \phi is the illumination.

7) Two lines before Equation 4: "Eq.(1)" must be "Eq.(2)"!!!!!

8) In the legend of Figure 2, the epsilon and alpha are not printed where they ought to be.

9) Line 282: "the physical meaning" should be "a physical meaning". And on the next line " than" should be ", then".

10) The sentence starting on line 286 is too long and ambiguous. It can be read as stating that "the number of activator maxima" must be higher than a threshold. In that case the 0.05 is very puzzling. It took me a while before I figured out what the author tries to say here. Just cut up the sentence and make it nonambiguous.

11) Line 293: "inhibition" should be "inhibitor".

12) Line 452: "for the for the"

13) Line 481: J_u must be J_v

14) Line 518: The values of alpha and beta were all identical? I am sure they were not all the same with alpha=beta. Be a bit more precise here and give the values. The idea should be that others can reproduce the numerical results.  

15) The format of the references is not consistent.

16) There are many instances where the use of articles "a" and "the" is erroneous or where they are missing where they should be.

Author Response

I am grateful for the positive evaluation of my manuscript and careful reading.

I think I have introduced all required corrections. The major modifications in the manuscript are typed in red in the second version of the manuscript. Many small language/style/typos corrections have been also introduced.